# Promotion of the Efficient Electrocatalytic Production of H_2_O_2_ by N,O- Co-Doped Porous Carbon

**DOI:** 10.3390/nano13071188

**Published:** 2023-03-27

**Authors:** Lina Sun, Liping Sun, Lihua Huo, Hui Zhao

**Affiliations:** 1Key Laboratory of Functional Inorganic Material Chemistry, Ministry of Education, School of Chemistry and Materials Science, Heilongjiang University, Harbin 150080, China; 2Key Laboratory of Molten Salts and Functional Materials of Heilongjiang Province, School of Science, Heihe University, Heihe 164300, China

**Keywords:** porous carbon, two-electron reduction of oxygen, hydrogen peroxide, electrocatalyst

## Abstract

H_2_O_2_ generation via an electrochemical two-electron oxygen reduction (2e^−^ ORR) is a potential candidate to replace the industrial anthraquinone process. In this study, porous carbon catalysts co-doped by nitrogen and oxygen are successfully synthesized by the pyrolysis and oxidation of a ZIF-67 precursor. The catalyst exhibits a selectivity of ~83.1% for 2e^−^ ORR, with the electron-transferring number approaching 2.33, and generation rate of 2909.79 mmol g^−1^ h^−1^ at 0.36 V (vs. RHE) in KOH solution (0.1 M). The results prove that graphitic N and –COOH functional groups act as the catalytic centers for this reaction, and the two functional groups work together to greatly enhance the performance of 2e^−^ ORR. In addition, the introduction of the –COOH functional group increases the hydrophilicity and the zeta potential of the carbon materials, which also promotes the 2e^−^ ORR. The study provides a new understanding of the production of H_2_O_2_ by electrocatalytic oxygen reduction with MOF-derived carbon catalysts.

## 1. Introduction

Hydrogen peroxide (H_2_O_2_) is regarded as one of the most significant chemical compounds [1,2,3]. It is serves as both a desirable energy carrier and a green oxidant and disinfectant. It is comprehensively applied in the industries of medicine and environmental protection, such as the bleach of paper pulp and textiles [4,5,6]. To date, the majority of the industrial processes for producing H_2_O_2_ depend on the process of anthraquinone oxidation. This is a complex and energy-intensive procedure, with a large quantity output of chemical waste [7,8,9]. Since Beal first reported the electrochemical production of H_2_O_2_ in the 1930s [10], the electrochemical reduction of oxygen has been gradually realized as a potential approach for the generation of H_2_O_2_ [11,12,13,14].

In order to produce H_2_O_2_ in an electrochemical way, it is crucial to explore novel electrocatalysts with superior activity, selectivity, stability and low cost [11]. Compared to noble metals and alloys (Pt-Hg [13], Pd-Au [15] and Pd-Hg [16]), carbon materials were extensively explored for their good abundance, low cost and easy functionalization [17,18,19,20,21,22,23]. The electronic structure of carbons is regulatable by some dopants such as heteroatoms [24,25], and the heteroatoms themselves can also act as active centers for ORR [26,27,28], thereby improving the catalytic activity. Particularly, it is of importance to dope nitrogen (N) in facilitating the oxygen reduction reaction through two electrons (2e^−^ ORR) for H_2_O_2_ production [29,30,31,32]. It is widely accepted that the doping of nitrogen into the carbon materials can drastically lower the overpotential of 2e^−^ ORR by the reduction in the Gibbs free-energy of O_2_ reduction and the optimization of the binding energy of HOO^−^ [31,33]. In alternative studies, the electrocatalytic activity of oxygen-substituted carbon materials is investigated. Xia et al. [34] explored a convenient approach to oxidize commercial carbon blacks with concentrated nitric acid. Oxygen functional groups were then introduced as active sites to increase the H_2_O_2_ selectivity up to ~95%. Recently, carbon materials with two kinds of active centers in the framework were prepared to further promote the catalytic property for H_2_O_2_ electroproduction. For example, Zhao [35] et al. prepared the COOH-terminated N-doping carbon aerogel. The production rate for H_2_O_2_ was as high as 0.071 mmol g^−1^ h^−1^, due to the synergy of the N atoms and the –COOH groups. Very recently, it was found that the significant interaction between the pyridinic N and carbon-based groups (–COOH/C–O–C) helped the desorption of intermediates *OOH, which subsequently enhanced the H_2_O_2_ selectivity, via the density functional theory (DFT) computation [36]. Therefore, the type and doping mode of dopants play crucial roles in electrochemical 2e^−^ ORR processes. In addition, the relationship between solvation effect and ORR selectivity has also been reported [37,38]. Therefore, it is necessary to investigate the hydrophilicity and charge of the catalyst.

Metal–organic framework materials (MOFs) have been widely considered as potential templates to produce carbon materials with adjustable heteroatoms, stable carbon skeleton structure, high specific surface area and abundant pore structures [39]. The carbon-based catalysts derived from MOFs are widely studied to be promising in ORR/OER (oxygen reduction reaction), HER (hydrogen evolution reaction) and CO_2_RR (CO_2_ reduction reaction) [40,41,42,43,44]. For the electrochemical production of H_2_O_2_, Sun [45] et al. and Gao [46] et al. demonstrated that MOF-derived Co–N–C single-atom catalysts (SACs) displayed good 2e^−^ ORR properties in acidic solutions, whereas the same Co–N–C SACs was found to show mainly 4e^−^ ORR performance in alkaline solutions [47,48,49,50]. In some other studies, MOF-derived carbon materials, such as layered porous carbon derived from MOF-5 [51], F-doped porous carbon prepared from MIL-53(Al) [28], NCPs synthesized by ZnPDA (zinc pyridine-2,6-dicarboxylate) [52] and ZIF-8-derived N-doped porous carbon *p*-ZIF [53], are considered for the electrochemical production of H_2_O_2_ in acidic solutions. Concerning the potential applications of H_2_O_2_ in industry, it is also interesting to produce H_2_O_2_ in alkaline solutions. In the present work, ZIF-67 (zeolitic imidazolate-67) was chosen as the precursor to synthesize porous carbon materials with a uniform distributed N dopant. The HNO_3_ (nitric acid) oxidation treatment was carried out to introduce –COOH, and at the same time to remove metal Co from the carbon framework. Finally, the carbon materials co-doped by N and –COOH were used as catalysts to investigate the effects of microstructure, dopant type, contact angle and Zeta potential on the property of 2e^−^ ORR. The ultra-high formation rate of 2909.79 mmol g^−1^ h^−1^ for H_2_O_2_ was reached in KOH solution (0.1 M), indicating that this material is a promising 2e^−^ ORR electrocatalyst.

## 2. Experimental

### 2.1. Reagents and Chemicals

2-Methylimidazole and cobalt nitrate hexahydrate (Co(NO_3_)_2_·6H_2_O) were provided from Aladdin Reagent Co., Ltd. (Shanghai, China); 5% Nafion solution was obtained from Aldrich chemical Co., Inc.(Du Pont, Wilmington, DE, USA). Nafion 115 membrane was obtained from Dupont. Carbon cloths (W0S1009) were purchased from Taiwan Carbon Energy Technology Co., Ltd.(Sinero Technology Co., Ltd. Suzhou, China). Methanol (≥99.5%), ethanol (≥99.5%), nitric acid (65%) and hydrochloric acid (36~38%) were provided from Sinopharm Group Chemical Reagent Co., Ltd. (Shanghai, China). All chemicals were employed without further treatment. 

### 2.2. Synthesis of Catalysts

*Synthesis of ZIF-67*: The dissolution of Co(NO_3_)_2_·6H_2_O (0.5850 g) and 2-methylimidazole (0.9850 g) was carried out in methanol (50 mL) to obtain two solutions. The 2-methylimidazole solution was charged drop-wise into the Co(NO_3_)_2_ solution with vigorous stirring. After stirring for 3 h at ambient temperature, the mixture was kept static for 12 h to obtain the sediment, followed by centrifuging, washing with ethanol and vacuum-drying at 70 °C for 12 h. The collected product was named ZIF-67.

*Synthesis of NPC-900*: A portion of ZIF-67 powder was heated in flowing N_2_ at 900 °C for 3 h. The carbonized product was then immersed in 15% concentration HCl solution, followed by stirring slowly for 12 h to eliminate the residual cobalt. After being washed with deionized water to pH = 7, the black powder obtained was vacuum-dried at 70 °C for 12 h. The collected product was named NPC-900.

*Synthesis of O-NPC-T*: In a typical pre-oxidation procedure, NPC-900 (50 mg) was charged into 100 mL of 65% nitric acid. The obtained solution was refluxed at 120 °C for 3 h, followed by washing to pH = 7. The black products were then vacuum-dried at 70 °C for 12 h and named as O-NPC-120. For comparison, the pre-oxidation was also performed at 80 °C and 100 °C, and the obtained products were named O-NPC-80 and O-NPC-100, respectively.

### 2.3. Characterization of Catalysts

The catalyst crystal structure was identified by X-ray diffraction (XRD, Bruker AXS D8: Bruker, Karlsruhe, Germany) under Cu Kα radiation. The morphologies and structures were observed by field emission scanning electron microscopy (FESEM, ZEIES-Sigma 500: ZEIES, Munich, Germany) and transmission electron microscopy (TEM, JEOL-JEM-2100, JEOL Ltd., Tokyo, Japan). Fourier-transformed infrared spectroscopy (FT-IR, PE spectroscopy ASCI: Bruker, Karlsruhe, Germany) was employed to study the formation of ZIF-67 precursor and the changes in chemical bonds during pyrolysis. X-ray photoelectron spectroscopy (XPS, KRATOS-AXIS-ULTRA-DLD: KRATOS, Manchester, UK) was used to obtain the surface composition of the catalyst. The C1s, O1s and N1s XPS spectra were analyzed with Casa XPS software version 2.3.23, with 284.6 eV as the charge-corrected reference for C 1. The degree of carbon defects was characterized by Raman spectroscopy (Raman, HORIBA-Lab RAM-HR: Jobin Yvon, Longjumeau, France). The specific surface area and pore size distribution of the catalysts were measured by the Micromeritics Instrument TriStar II 3020 ( Micromeritics Instrument, Norcross, GA, USA) for N_2_ adsorption-analytical isotherms, on the basis of the Brunauer–Emmett–Teller (BET) equation and the Barrett–Joyner–Halenda (BJH) method. UV-Vis spectrum was collected on Shimadzu UV-2600 UV-Vis spectrophotometer (Shimadzu, Kyoto, Japan). The Zeta potential of the material surface in aqueous solution was determined using Malvern Zetasizer Nano ZS90 (Malvern Instruments Ltd., Malvern, UK). The content of metallic Co in the material was determined using ICP-MS Aglient 7800 (Agilent, Santa Clara, CA, USA).

### 2.4. Electrochemical Measurements

An electrochemical workstation (CHI 760 B) and Pine Rotator (Instrument model: AFMSRCE, Pine Research Instrumentation, Inc., Durham, NC, USA) were employed to explore the catalytic properties of the electrocatalysts at 25 °C. All the electrochemical characterization was performed in a standardized three-electrode cell. An RRDE (RDE: 0.2475 cm^2^, Pt ring: 0.1866 cm^2^) loaded with catalysts was taken as a working electrode, and a Pt-mesh and saturated Hg/HgO (1 M KOH) electrode were taken as a counter electrode and a reference electrode, respectively.

Preparation of RRDE working electrode: The dispersion of the catalyst (5 mg) in an aqueous solution with 1960 μL of isopropanol solution ($V_{\mathrm{isopropanol}}{:V}_{H_{2}O}=1:3$) and Nafion (40 μL, 5 wt.%) was conducted under 2 h of sonication, to form the homogeneous catalyst ink. The ink (10 μL) was then transferred onto the RRDE surface, followed by drying at ambient temperature in air to synthesize the working electrode. Additionally, the catalyst loading amount was 100 μg/cm^2^.

The measurements of cyclic voltammetry (CV) and linear sweep voltammetry (LSV) were carried out from 0 to 1.0 V (vs. *RHE*) in N_2_- or O_2_-saturated 0.1 M KOH electrolytes (pH 13) at a scanning speed of 50 mV s^−1^ and 10 mV s^−1^, respectively. The rotation speed of RRDE was 1600 rpm. The ring potential was set at 1.2 V (vs. *RHE*) to quantitatively detect the H_2_O_2_. The conversion of all the potentials into reversible hydrogen electrodes (RHE) was performed by the subsequent equation:(1)$$E_{RHE\;}=E_{HgO}+0.0591\times pH+0.098\;(V)$$

The electron transfer number (*n*), the H_2_O_2_ selectivity (H_2_O_2_%) and the Tafel slope η are computed from the RRDE polarization curve with the subsequent equations:(2)$$n=\frac{4\times N\times \left|I_{disk}\right|}{N\times \left|I_{disk}\right|+I_{ring}}$$
(3)$$H_{2}O_{2}\%=\frac{200\times I_{ring}}{N\;\times \left|I_{disk}\right|+I_{ring}}$$
(4)$$\eta =b\;\log\;j_{k}+a$$
where *I_disk_* is the disk current (mA cm^−2^), *I_ring_* is the ring current (mA cm^−2^). *N* is the collection efficiency of Pt ring, which is calculated to be 0.35 (the collection efficiency is obtained with a one-electron reversible ${\left[\mathrm{Fe}(\mathrm{CN}\right)}_{6}^{4-}]/[{\mathrm{Fe}(\mathrm{CN})}_{6}^{3-}]$ system in K_3_[Fe(CN)_6_] solution). *j* indicates the kinetic current density and b refers to the Tafel slope, $j_{k}$ is kinetic current (mA cm^−2^). 

According to the K-L equation, the kinetic current density is calculated as follows:(5)$$\begin{array}{l} \frac{1}{j}=\frac{1}{j_{k}}+\;\frac{1}{j_{L}} \end{array}$$
(6)$$j_{L}=0.62nFD_{O_{2}}^{\frac{2}{3}}\upsilon^{-\frac{1}{6}}\omega^{\frac{1}{2}}C_{O_{2}}$$
where *j* is the measured current density and $j_{k}$ and $j_{L}$ are the kinetic current and diffusion-limited current densities, respectively. *n* is the number of electrons transferred, *F* is the Faraday constant (96,485 C mol^−1^), $D_{O_{2}}$ is the diffusion coefficient of oxygen (1.9 × 10^−5^ cm^2^ s^−1^), õ is the kinematic viscosity of the solution (0.01 cm^2^ s^−1^), *ù* is the angular velocity (in rpm), $C_{O_{2}}$ is the bulk concentration of O_2_ (1.2 × 10^−3^ mol L^−1^).

### 2.5. Determination of H_2_O_2_ Production and Faradaic Efficiency

In order to eliminate the occurrence of the reaction of the H_2_O_2_ generated on the counter electrode, the tests were performed in an H−type electrolytic cell equipped with a pretreated Nafion 115 separator. The specific test process was as follows: the catalyst ink was coated on commercial carbon cloth (1 × 1 cm^2^) and dried at ambient temperature to prepare the working electrode. The cell was filled with KOH solution (40 mL, 0.1 M). Before testing, the cathode compartment was purged with a high-purity oxygen gas for no less than 30 min, and oxygen was kept continuously supplied to the cathode compartment. In situ electrolysis was performed at 0.26 V, 0.36 V and 0.46 V by chronoamperometry for 3 h. A certain amount of electrolyte was taken every 30 min, and the quantification of generated H_2_O_2_ (HO_2_^−^) was performed by the Ce (SO_4_)_2_ titration approach.
(7)$$2{\;\mathrm{Ce}}^{4+}{+\;H}_{2}O_{2}\;\to {\;2\;\mathrm{Ce}}^{3+}{+\;2\;H}^{+}{+\;O}_{2}\;↑$$

According to the concentration of reduced Ce^4+^, the Faradaic efficiency (*FE%*) for H_2_O_2_ formation can be obtained with the subsequent formula:(8)$$FE\%=\frac{Q_{H_{2}O_{2}}}{Q_{total}}\times \;100\%=\frac{2\;C_{H_{2}O_{2}}\;V\times F}{Q_{total}}\times \;100\%$$
where $Q_{H_{2}O_{2}}$ is the charge consumed to produce H_2_O_2_ (C), $Q_{total}$ represents the total charge (C) passed in the chronoamperometric test in 3 h, which was realized by the subtraction of the charges measured in nitrogen-saturated solution from those in oxygen-saturated one, $C_{H_{2}O_{2}}$ refers to the concentration of H_2_O_2_ produced (mol L^−1^), *V* is the volume of electrolyte (L), *i* denotes the current during electrolysis (A), *F* indicates Faraday’s constant (96,485 C/mol) and *t* is the electrolysis time (s).

## 3. Results and Discussion

### 3.1. Synthesis and Characterization of Catalysts 

Figure 1a presents the XRD patterns of the prepared material. Clearly all the diffraction peaks are located exactly in the same positions as published in the literature [54,55], proving the successful formation of single-phase ZIF-67. Figure 1b shows the XRD pattern of the pyrolysis product and the oxidation products that were pre-treated by concentrated nitric acid at different temperatures. All these materials show diffraction peaks at 25.99°, 42.66°, 53.62° and 77.66°, attributed to the (002), (100), (004) and (110) planes of graphitic carbon, respectively. In the pyrolysis product, the small diffraction peaks at 44.18°, 51.56° and 75.78° correspond to the (111), (200) and (220) crystal planes of metallic Co (JCPDS: 15-0806). The intensities of these diffraction peaks gradually decrease with the increase in pre-oxidation temperatures. At 100 °C and 120 °C, these diffraction peaks disappear, indicating that the Co element in the material is completely removed after the nitrate acid pre-oxidation treatment. To further study whether there is residual cobalt in the catalyst, the content of metallic Co in the catalyst material is measured by ICP-MS, and the result is shown in Table 1. The Co content decreases with the rising temperature of pre-oxidation. When the pre-oxidation temperature reaches 100 °C and 120 °C, the content of metallic Co is less than the detection limit (0.01%). This result is consistent with the EDS data (0.03%, Appendix A), indicating almost no metallic Co in the catalyst.

Figure 1c shows the FT-IR spectra of corresponding studied materials. For ZIF-67, the typical FT-IR spectrum is obtained [55,56,57,58]. The peak at 3425 cm^−1^ is attributed to the –OH stretching vibration; 1580 cm^−1^ is due to the C=C stretching vibration; the peaks at 1417 cm^−1^ and 1303 cm^−1^ can be assigned to the C=N and C–N stretching vibrations. The other two peaks at 1141 cm^−1^ and 755 cm^−1^ belong to the C–H and C=N bending vibrations. In summary, all the peaks are derived from the vibrations of the imidazole ring. In addition, the peak at 425 cm^−1^ refers to the vibration of Co–N coordination bonds. Therefore, combined with the XRD characterizations, it shows clearly that ZIF-67 is successfully synthesized. After the pyrolysis treatment at 900 °C, the vibration peaks of –OH and the imidazole ring disappear, indicating the decomposition of the ZIF-67 precursor to form carbon-related products. After the pre-oxidation treatment, two newly developed peaks at ~1700 cm^−1^ and ~1205 cm^−1^ are attributable to the stretching vibration of C=O and C–O, showing the successful integration of O–containing functional groups into the pyrolysis products. 

Carbon defect is a significant indicator to determine the catalytic ability of carbons for H_2_O_2_ formation via the 2e^−^ ORR pathway [51,59]. To characterize the possible carbon defects in our material, Raman spectrum is collected (Figure 1d). The D-band and G-band appear at ~1350 cm^−1^ and ~1600 cm^−1^ in the four studied materials. Typically, the intensity ratio of D-band to G-band (I_D_/I_G_) describes the relative defect contents in carbon materials, mainly due to the fact that D-band and G-band are associated with the disordered and ordered crystalline sp^2^-C, respectively [23]. It is generally believed that the larger the ratio, the higher the amounts of defects. However, it is found from Figure 1d that the I_D_/I_G_ are 0.99, 0.99, 1.01 and 1.00 for NPC-900, O-NPC-80, O-NPC-100 and O-NPC-120, respectively. This result indicates that the relative carbon defect contents are quite similar in the four studied carbon materials. The specific surface area and porosity of the material are further calculated by N_2_-sorption isotherms. At the relative pressure of 0.4 < p/p_0_ < 1.0, the hysteresis loops are found in all samples, indicating the generation of mesoporous structures (Figure 1e). The curves of pore diameter distribution verify the existence of a mesoporous structure with apparent peaks of around 2.3~4.3 nm (Figure 1f). Meanwhile, stacking pores with pore size around 5.5~10 nm are also observed in NPC-900 and O-NPC-80. The calculated pore volumes of NPC-900 (0.4913 cm^3^/g) and O-NPC-80 (0.4196 cm^3^/g) are significantly larger than those of O-NPC-100 (0.3822 cm^3^/g) and O-NPC-120 (0.3223 cm^3^/g). Furthermore, the Brunauer–Emmett–Teller (BET) surface areas of NPC-900, O-NPC-80, O-NPC-100 and O-NPC-120 are 226.40 m^2^/g, 157.65 m^2^/g, 92.11 m^2^/g and 56.58 m^2^/g, respectively. The pre-oxidation temperature increases with the gradually decreasing specific surface area of the material, probably due to the collapse of the pore structure. This conclusion can also be drawn from the observation of the microstructure evolution of the catalysts (Appendix A).

### 3.2. 2e^−^ ORR Performance of Catalysts

The half-wave potential (*E*_1/2_) is a key index to assess the ORR electrocatalyst activity [60]. Based on the linear sweep voltammetry plots, the *E*_1/2_ of NPC-900 is 0.78 V, while it is 0.74 V, 0.75 V and 0.73 V for O-NPC-80, O-NPC-100 and O-NPC-120, respectively (Figure 2a). The *E*_1/2_ of O-NPC-120 is very similar to the thermodynamic potential of 2e^−^ ORR (≈0.7 V) [56,58], indicating that it has the best 2e^−^ ORR activity. Meanwhile, the ring current of O-NPC-120 reaches 1.80 mA cm^−2^, which is the highest one among the four studied materials. The CV curves of the catalysts indicate obvious ORR catalytic activity of these carbon materials (Appendix A). Figure 2b,c show the electron transfer number and selectivity of H_2_O_2_ that were obtained according to the plots of Figure 2a. Clearly, O-NPC-120 shows the best performance, with *n* = 2.33, and the selectivity reaches 83.10%. The electron transfer number was further calculated through the K-L equation (Appendix A), and the result was found to be consistent with the RRDE measurement. To gain a further understanding of the 2e^−^ ORR performance, the Tafel slope values are calculated and the results are presented in Figure 2d. It is found that the values are between 35 and 47 mV dec^−1^ for the four studied materials. This value is smaller than some carbon catalysts reported in the literature [23,61,62,63,64,65,66], indicating that ZIF-67-derived carbon has much faster kinetics for the 2e^−^ ORR reaction. The effects of catalyst loading are studied and the results are presented in Appendix A. Clearly, when the catalyst loading is 100 ìg/cm^2^, the electrode shows the best 2e^−^ ORR performance. Figure 2e shows the stability test for 10 h. Both the ring current and the disk current have no obvious attenuation after 10 h consecutive recording, indicating the quite super stability of the catalyst. The ORR performance of the glassy carbon electrode (GCE) without the loading of catalysts was evaluated and the results are comparably shown in Appendix A. This result indicates that the major contribution of H_2_O_2_ production comes from the catalytic activity of O-NPC-120.

To further understand the important parameters determining the 2e^−^ ORR activity and selectivity, the electrochemical double-layer capacitance (C_dl_) of the four carbon materials is tested. According to Figure 2g, the C_dl_ of NPC-900, O-NPC-80, O-NPC-100 and O-NPC-120 is 2.62 mF cm^−2^, 2.12 mF cm^−2^, 1.52 mF cm^−2^ and 1.35 mF cm^−2^, respectively. Considering that the electrochemical active surface area (ECSA) is positively proportional to C_dl_ [67,68], this result shows that O-NPC-120 has the smallest ECSA among the four materials. In addition, the Raman analysis has proved that there is no significant difference of carbon defect contents in the four studied carbon materials (Figure 1d). 

### 3.3. Production Test of H_2_O_2_

In order to obtain the working potential for the formation of H_2_O_2_, and to eliminate the reduction of the generated H_2_O_2_ during the oxygen reduction process, the reduction reaction test of H_2_O_2_ was carried out (Appendix A). The currents that appear over 0.79 V and below 0.25 V are assigned to the currents of oxidation and reduction of H_2_O_2_ [29], respectively. Therefore, the applied voltages of 0.26 V, 0.36 V and 0.46 V were selected in the H_2_O_2_ production rate measurement experiments. The amount of H_2_O_2_ produced is calculated by the spectroscopic results (Appendix A), which are normalized by electrolysis time and catalyst loading to obtain the H_2_O_2_ production rate (Figure 3a). It is observed that the H_2_O_2_ amount increases gradually with the electrolysis time. The O-NPC-120 electrocatalyst exhibits a high H_2_O_2_ production rate of 2907.79 mmol g_catalyst_^−1^ h^−1^ at 0.36 V, significantly larger than the H_2_O_2_ production rate reported [69,70,71,72,73,74,75] (Appendix A). Figure 3b is the faradaic efficiency diagram of O-NPC-120. The Faradaic efficiency reaches 95.63% at 0.36 V, much better than that at 0.26 V and 0.46 V.

### 3.4. Catalytic Mechanism Analysis

Hydrophilicity is a critical feature for H_2_O_2_ generation [66]. To examine the wetting ability of the catalysts to the 0.1 M KOH electrolyte solution, contact angle tests were performed (Figure 4). The contact angles of NPC-900, O-NPC-80, O-NPC-100 and O-NPC-120 are 143.5°, 73.5°, 41.3° and 24.0°. This result shows that the contact angle decreases dramatically with the pre-oxidation treatment temperatures. This is attributed to the integration of O-containing functional groups, which increases the hydrophilicity of the carbon material. The good hydrophilicity may facilitate the mutual interaction between the carbon catalyst and the electrolyte, and contribute to the diffusion of O_2_.

Zeta potential is another important factor affecting ORR [76]. The Zeta potentials of NPC-900, O-NPC-80, O-NPC-100 and O-NPC-120 are measured to be −6.04 mV, −29.6 mV, −39.4 mV and −40.4 mV, respectively (Figure 5a). It can be seen that Zeta potential becomes more negative with the increase in pre-oxidation treatment temperatures, with O-NPC-120 showing the most negative Zeta value. This result indicates that O-NPC-120 has the strongest desorption ability for the adsorbed intermediate species OOH^−^, due to the coulombic repulsion effects of this carbon material to negatively charged species. It is well established in the literature that the easy desorption of OOH^−^ from the catalyst surface is beneficial for the 2e^−^ ORR reaction, therefore O-NPC-120 shows the best catalytic performance for H_2_O_2_ production.

The chemical components and bonding states are characterized by XPS, so as to determine the active centres of the catalyst (Figure 5b–d and Appendix A). Clearly, the strong signals of C1s, N1s and O1s are found at ~285.0 eV, ~401.0 eV and ~531.6 eV, respectively (the specific contents of C, N and O are shown in Appendix A). The C1s signal of O-NPC-120 can be decomposed into five types (Figure 5b), namely, sp^2^-C: C=C (284.60 eV), sp^3^-C: C–C (285.06 eV), C–O/C–N (286.00 eV), –COOH (288.89 eV) and π→π* (290.61 eV). The corresponding atomic percentages are 32.01%, 25.29%, 25.1%, 5.95% and 11.65% (Appendix A). The O1s signal is decomposed into three types (Figure 5c), namely, C=O (531.76 eV), C–O–C/–OH (532.40 eV) and –COOH (533.38 eV), and their relative concentrations are 43.50%, 25.43% and 31.07% (Appendix A). In summary, the successful integration of –COOH into the carbon framework is realized after the nitric acid pre-oxidation treatment. It is also observed that the content of –COOH increases with the increase in pre-oxidation temperatures (Appendix A). The N1s signal at ~400 eV can be decomposed into three types (Figure 5d), namely, pyridinic N (398.54 eV), pyrrolic N (400.56 eV) and graphitic N (401.63 eV). Another peak at 405.80 eV is due to the formation of N–oxide. The corresponding atomic percentages of various N species are shown in Appendix A. In order to understand the contributions of different N– and O–containing species to the catalytic activity, the variations of H_2_O_2_ selectivity versus atomic percentages of O and N species are plotted in Figure 5e,f, respectively. Clearly, the selectivity of H_2_O_2_ increases with the increase in –COOH and graphitic N contents. Therefore, we propose that both the –COOH and graphitic N are the catalytic centres for the 2e^−^ ORR reaction. Furthermore, DFT simulation results have confirmed the improvement effects of the coupled N/COOH complexes on the improved adsorption of O_2_ [35,77,78]. Moreover, the nitrogen-based and COOH−based groups may also play the roles of the intramolecular acid/base to aid the catalytic reactions [79,80]. Therefore, we believe that the excellent performance of O-NPC-120 is due to the joint contributions of graphite N and –COOH.

At last, the active site density is calculated according to the following formula [68]:(9)$$\begin{array}{c} \mathrm{ASSD}=\frac{m_{catalyst}\times O\%(at\%)\times (-COOH\%)(at\%)\;}{m_{catalyst}\times BET(m^{2}{\;g}^{-1})}+\frac{m_{catalyst}\times N\%(at\%)\times (Graphtic\;N\%)(at\%)}{m_{catalyst}\times BET(m^{2}{\;g}^{-1})}\\ =\left[\frac{O\%(at\%)\times (-COOH\%)(at\%)}{BET}+\frac{N\%(at\%)\times (Graphtic\;N\%)(at\%)}{BET}\right]\times 1000\;({\mathrm{mg}\;m}^{-2}) \end{array}$$

$m_{catalyst}$ is the mass of catalyst (g); O%(at%) and N%(at%) are the content of O1s and N1s characterized by XPS; –COOH%(at%) and graphitic N%(at%) are the content of –COOH and graphitic N obtained by XPS deconvolution of O1s and N1s.

It can be seen from Table 2 that with the increasing pre-oxidation treatment temperature, the active site density gradually increases, which also makes the activity and selectivity of 2e^−^ ORR increase sequentially.

## 4. Conclusions

ZIF-67 was used as the precursor, and the oxygen-containing functional group (–COOH) was successfully introduced into the carbon skeleton through high-temperature carbonization and a concentrated nitric acid oxidation reaction. The results show that the electron transfer number of O-NPC-120 is 2.33, and the selectivity of H_2_O_2_ is 83.10%. The high H_2_O_2_ formation rate of −2909.79 mmol g _catalist_^−1^ h^−1^ was obtained with O-NPC-120 at 0.36 V. The superior property of this catalyst is mostly due to the following aspects: (1) Thegraphitic N and –COOH functional groups act as catalytic sites, and they work together to greatly enhance the performance of 2e^−^ ORR. (2) The catalyst has good hydrophilicity, which can promote the mutual contact between the catalyst and electrolyte and contribute to the diffusion of O_2_. (3) The catalyst has the largest negative Zeta potential value, which is of benefit to the desorption of adsorbed intermediate OOH^−^. The present work is expected to be helpful to rationally design efficient carbon-based catalysts for the production of H_2_O_2_.

## Figures and Tables

**Figure 1 nanomaterials-13-01188-f001:**
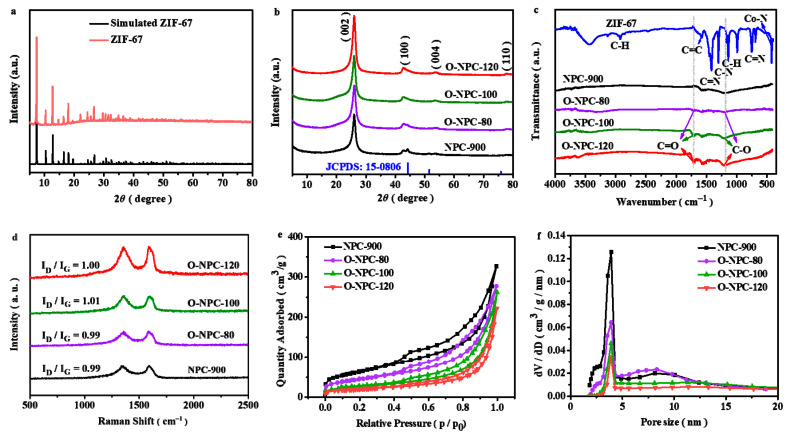
XRD patterns of (**a**) the precursor ZIF-67. (**b**) NPC-900 and O-NPC-T. (**c**) FT-IR spectra of ZIF-67, NPC-900 and O-NPC-T. (**d**) Raman spectra of NPC-900 and O-NPC-T. (**e**) N adsorption–desorption isotherms of NPC-900 and O-NPC-T at 77K. (**f**) Pore size distribution of NPC-900 and O-NPC-T.

**Figure 2 nanomaterials-13-01188-f002:**
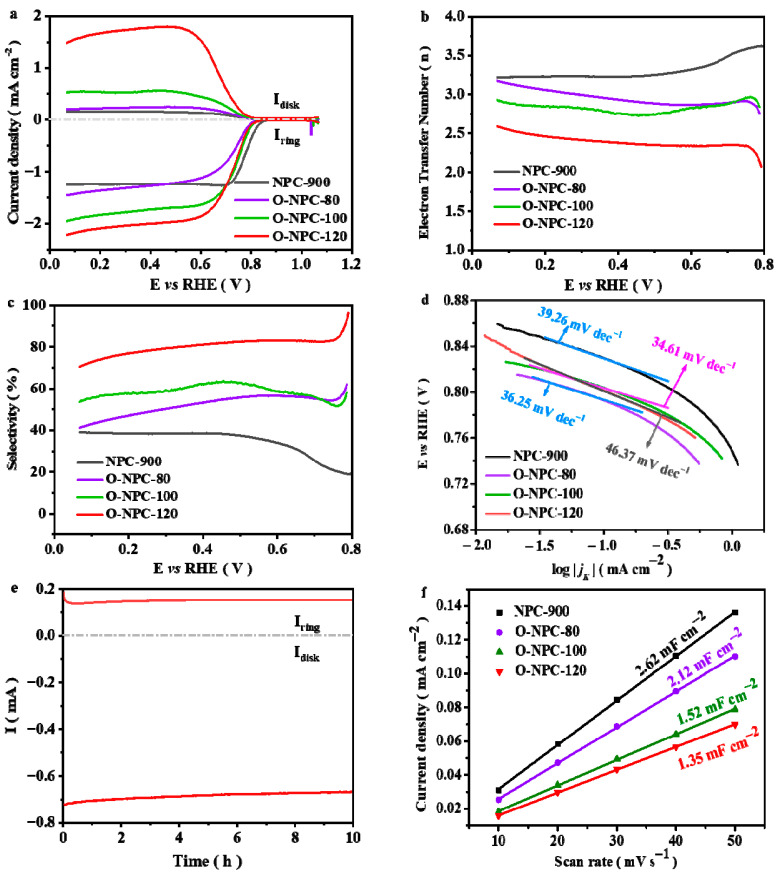
The catalyst ORR property in KOH electrolyte (0.1 M): (**a**) The catalyst RRDE polarization curves at 1600 rpm and a scan rate of 10 mV s^−1^. Current density was normalized to loading and BET. (**b**) Electron transfer number and (**c**) H_2_O_2_ selectivity of catalysts. (**d**) Tafel slope of catalysts. (**e**) Stability test of O-NPC-120 at a fixed disk potential of 0.36 V. (**f**) The capacitance current densities (obtained from Appendix A) tested at 0.963 V (vs. RHE) as a function of scanning rate. The slope value refers to the double-layer capacitance (C_dl_).

**Figure 3 nanomaterials-13-01188-f003:**
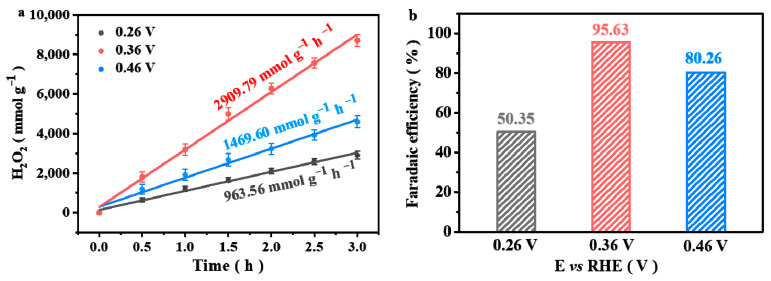
(**a**) H_2_O_2_ production rate; (**b**) the corresponding faradaic efficiency (FE%).

**Figure 4 nanomaterials-13-01188-f004:**
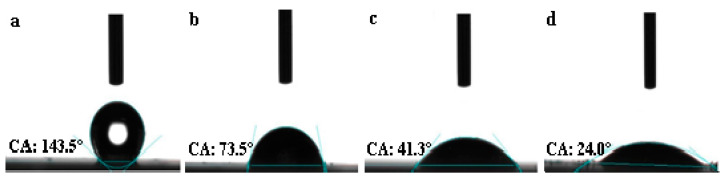
Contact angle pictures of catalysts: (**a**) NPC-900. (**b**) O-NPC-80. (**c**) O-NPC-100. (**d**) O-NPC-120.

**Figure 5 nanomaterials-13-01188-f005:**
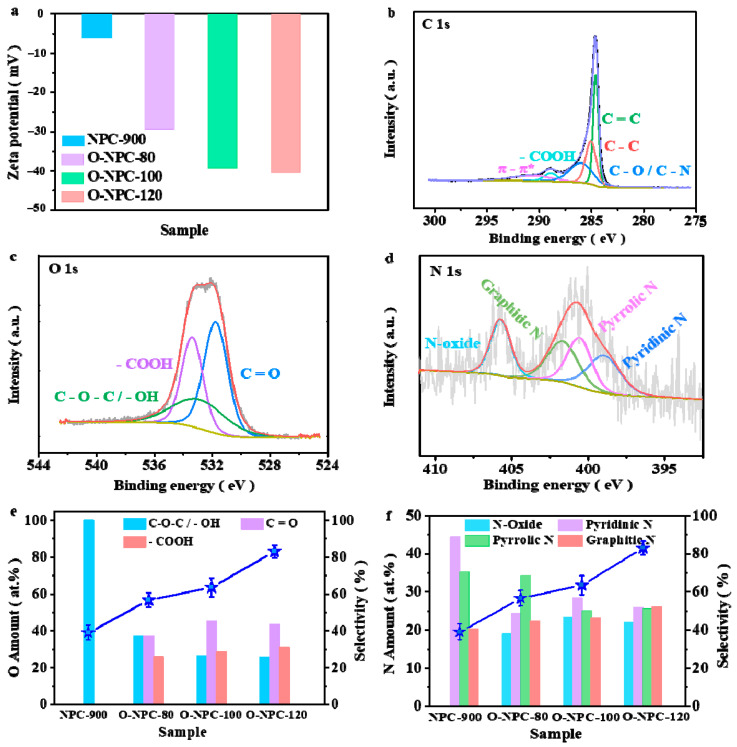
(**a**) Zeta potential of catalysts in water (pH = 7). XPS spectra of O-NPC-120 (**b**) C1s, (**c**) O1s, (**d**) N1s. (**e**) A plot of the relationship between the atomic ratio of O and H_2_O_2_ selectivity. (**f**) A plot of the relationship between the atomic ratio of N and H_2_O_2_ selectivity.

**Table 1 nanomaterials-13-01188-t001:** Determination of Co content in catalysts by ICP-MS.

Sample	NPC-900	O-NPC-80	O-NPC-100	O-NPC-120
Co (wt %)	0.89	0.16	<0.01	<0.01

**Table 2 nanomaterials-13-01188-t002:** Calculation results of active site density.

Sample	NPC-900	O-NPC-80	O-NPC-100	O-NPC-120
*ASSD*	0.0112	0.2036	0.4202	0.8449

## Data Availability

The data presented in this study are available on request from the corresponding author.

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
