# Peer review of "Promotion of the Efficient Electrocatalytic Production of H2O2 by N,O- Co-Doped Porous Carbon"

_nanomaterials, 2023, doi:10.3390/nano13071188_

Round 1

Reviewer 1 Report

In this paper, the authors introduced a new catalyst for hydrogen peroxides production in alkaline media, synthesized by the pyrolysis and oxidation of MOF derived carbon catalysts.

Based on spectroscopic and electrochemical methods, the authors proposed that the co-existed Graphitic N and - COOH functional groups act as active sites for the reaction, and the synergistic effect between the two groups greatly enhances peroxides anion production. The results tied the hydrophilicity between the catalyst and electrolyte to the diffusion of O2, and the Zeta potential value to the adsorbed intermediate OOH-. Some minor revisions are needed:

1.     The amount of hydrogen peroxide produced is proportional to the amount of electrocatalyst on the electrode, as its well-documented in the literature. From this point of view, the study performed should be repeated at a different catalyst loadings to confirm the conclusions obtained.

2.     The authors use several methods to prove the peroxides production mechanism. It is common to use the Levich equation to calculate the electron number. This data could strengthen the mechanism suggested in the paper.

3.     In lines 282-283 the authors mention “The current appears above 0.79 V and below 0.25 V are attributed to H2O2 oxidation current and reduction current, respectively...” There should be further explanation for the choice of the these working potentials. Why isn’t there a greater variety between the range of 0.25V-0.79V?

4.     The compounds of the various catalysts are thoroughly studied in both the article and SI. To find out in what oxidation state the residual cobalt is present in the samples it is worthwhile to add the XPS survey.

5.     The authors discuss the importance of pore size distribution in lines 225–230, however the distribution of the pores is not mentioned anywhere in the text. Additionally, it is understood from the literature that the wetting ability within the pores differs from that found at the surface. How is this phenomenon interacting with the wetting angle results?

6.     The authors should refer to the work in the following studies in their introduction.

a) Noffke, B. W., Li, Q., Raghavachari, K., & Li, L. S. (2016). A model for the pH-dependent selectivity of the oxygen reduction reaction electrocatalyzed by N-doped graphitic carbon. Journal of the American Chemical Society138(42), 13923-13929.

b) Honig, H. C., Krishnamurthy, C. B., Borge-Durán, I., Tasior, M., Gryko, D. T., Grinberg, I., & Elbaz, L. (2019). Structural and physical parameters controlling the oxygen reduction reaction selectivity with carboxylic acid-substituted cobalt corroles incorporated in a porous carbon support. The Journal of Physical Chemistry C123(43), 26351-26357.

Reviewer 2 Report

The authors study the N,O-doped porous carbon prepared from ZIF-67 as electrocatalyst for H2O2 production. The study is interesting, and I only have some minor suggestions to improve the manuscript.

I suggest to recalculate numbers to have same units as this will ease the comparison between literature studies and the current study. As example: In line 51 the unit is mg L-1 g-1 h-1, but in line 76 the unit is mmol g-1 h-1.

Something is missing in a bracket in lin 132.

In Table 1, the > should be < as the Co-content is less than (<) 0.01.

In line 255, the authors refer to Table S6, this should be S5.

At what pH was the zeta potential measured?

Is the concentration of H2O2 based on the equation from the standard curve in Fig. S6? If yes, why is the linerized plot not forced through 0,0 as the absorbance should be 0 at 0 mM Ce4+? Please justify.

Reviewer 3 Report

This work by Zhao et al describes the synthesis of several carbon catalysts by pyrolysis of ZIF-65 and oxidation at different T. The obtained samples were tested as catalysts in the 2e-ORR to produce hydrogen peroxide. Extensive characterization of each carbon catalyst was also performed to investigate the reason for the outstanding activity of the sample O-NPC-120.

The paper contains a well-structured study but with several weak points. In my opinion this article needs some major reconsiderations that I list below:

- Some acronyms are not explained. For instance for Zn-PDA, OER and HER in the Introduction or DI (line 102) it is not written what the acronym stands for.

- Reference 14 is repeated in line 32 and 35 in different contexts. A reference for Pt-Hg alloys as ORR catalysts is therefore missing.

- In my opinion, the use of the integral in formula 6 is inappropriate

- each catalytic test is performed with an electrode on which the catalyst suspension was absorbed. This operation may be not perfectly reproducible. Did the authors do replicates for the H2O2 production tests? Can they provide error bars for the data pictured in Figure 3 and for the selectivity values in figure 5e/5f? This should be important to confirm the high productivity of O-NPC-120

- carboxylic groups are found in the XPS analysis and are thought to be active sites for the ORR. However, in the IR spectra of the carbon catalysts it is hard to find signals corresponding to the OH bending. Can the authors comment on this?

- I think formula 7 should be better explained in the text, the meaning of some factors is not clear

- Saying "ultra-high productivity" for  O-NPC-120 seems inappropriate. At first, probably many metal catalysts can outperform this result. Secondly, O-NPC-120 seems not to be the best among the carbon based catalysts. In Table S5 the productivity of some reported carbon catalysts for ORR is shown in different unit misures and it is hard to compare. For instance, the productivity for Reference 5 in the supporting information is probably higher than the one in the present paper, since it is not divided by the catalyst weight. I think the result of 2786.58 mmol g catalist-1 h-1 is good but should be put in perspective.

- the authors state in line 277 that the number of defects in the carbon framework is not a key factor. Since all the studied samples had a similar amount of defects, this statement seems unjustified, as there is no information on the changes the catalytic activity depending on the defect content.

- I don't think the authors proved the existence of a sinergy between the COOH and the graphitic N sites. By the percentages of COOH and graphitic N in figure 5e-5f, the COOH/N ratio does not seem to change significantly in each sample, while their H2O2 selectivities are different. Therefore, I don't undestand on what basis the authors propose such sinergy.

Reviewer 4 Report

The manuscript by Sun Lina et al. represents the investigation of the electrocatalysis of oxygen reduction reaction on carbon materials. The obtained data is reliable and important. I would recommend major revision in order to achieve the standardization on RRDE data.

1.      The RRDE curves for empty glassy carbon should be present because it has an ORR-to-H2O2 activity in alkaline media.

2.      It would be nice to present the potential dependence of mass-normalized kinetic current obtained from K-L study (especially at the potentials 0.75-0.6 V (RHE)). Also, the normalization of kinetic currents by capacitive currents (or just by BET surface area) would be helpful to present.

Ideally, it would be nice to present the potential dependence of mass-normalized kinetic currents for different mass loadings. However, this point can be considered as a suggestion for future measurements.

Round 2

Reviewer 1 Report

Thank for answering all of my comments. I have no further comments and recommend to publish this manuscript in its current form.

Author Response

We thank the reviewer very much for the positive comment of our revised manuscript

Reviewer 3 Report

I think the authors answered properly to most of the issues I raised in the previous review and I thank them to have considered my suggestions. I have just a couple of observations related to two points previously raised:

Review 1: - 5. -carboxylic groups are found in the XPS analysis and are thought to be active sites
for the ORR. However, in the IR spectra of the carbon catalysts it is hard to find

signals corresponding to the OH bending. Can the authors comment on this?

Answer
: Thanks for this valuable comments. We agree that it is hard to find signals
corresponding to the OH bending in Fig. 1c. Therefore the figure has been enlarged in

the wavenumber range from 2000cm
-1 to 1000cm-1, to show the details of the
corresponding FT
-IR spectrum. Some broad peaks could be observed in the
wavenumber of 1600cm
-1 to 1450cm-1, and the peak intensity increases gradually
with the pre
-oxidation temperatures. It is known that the vibration bands of
1490~1350cm
-1 are attributed to the bending vibration in CH plane. The tensile
vibration
of C=C is ~1500cm-1; The in-plane bending vibration peak of -OH is at
~1450cm
-1. Due to the serious overlap of these peaks, it is hard to find signals
corresponding to the OH bending

Review 2: I apologize, I wrote OH bending by mistake. I was referring to O-H stretching signals, which for the COOH should be a strong signal around 3000 cm-1. Can the authors comment about the lack of these signals?

Review 1: 7. - Saying "ultra-high productivity" for O-NPC-120 seems inappropriate. At first,
probably many metal catalysts can outperform th
is result. Secondly, O-NPC-120
seems not to be the best among the carbon based catalysts. In Table S5 the

productivity of some reported carbon catalysts for ORR is shown in different unit

misures and it is hard to compare. For instance, the productivity fo
r Reference 5 in
the supporting information is probably higher than the one in the present paper, since

it is not divided by the catalyst weight. I think the result of 2786.58 mmol g catalist
-1
h
-
1 is good but should be put in perspective.
Answer
: We thank the reviewer for this valuable comment to improve the clarity of
our manuscript. According to the comment, modifications have been made in the

Abstract( line 17), Results(line 314, 316) and Conclusion(line 395)

Review 2: I thank the authors for considering my suggestions. However, the term "ultra-high" is still present in the conclusions.

The productivity value in Table S5 in the same unit of measure are easier to compare. I have some doubts concerning the productivity value for refence 5 in the SI, in which it was reported a productivity of hydrogen peroxide of 1,975 mg l−1 within 30 min by using a 25 ml cell and about 0.02 mg of catalyst. Therefore, their productivity should be 14500 mmol gcatalyst-1 h-1 if I undestood correctly. The authors wrote the productivity of reference 5 as 2323.5 mmol g-1 h-1 . Can the authors clarify how they did the conversion?

Reviewer 4 Report

The revised manuscript can be accepted for publication.

Author Response

(The authors gave the same response as above.)
